# *Plasmodium falciparum* Malaria Vaccines and Vaccine Adjuvants

**DOI:** 10.3390/vaccines9101072

**Published:** 2021-09-24

**Authors:** Srinivasa Reddy Bonam, Laurent Rénia, Ganesh Tadepalli, Jagadeesh Bayry, Halmuthur Mahabalarao Sampath Kumar

**Affiliations:** 1Institut National de la Santé et de la Recherche Médicale, Centre de Recherche des Cordeliers, Equipe-Immunopathologie et Immunointervention Thérapeutique, Sorbonne Université, Université de Paris, F-75006 Paris, France; jagadeesh.bayry@crc.jussieu.fr; 2A*STAR Infectious Diseases Labs, 8A Biomedical Grove, Singapore 138648, Singapore; 3Lee Kong Chian School of Medicine, Nanyang Technological University, Singapore 308232, Singapore; 4School of Biological Sciences, Nanyang Technological University, Singapore 308232, Singapore; 5Vaccine Immunology Laboratory, Organic Synthesis and Process Chemistry Division, CSIR-Indian Institute of Chemical Technology, Hyderabad 500007, India; ganeshbabutmln@gmail.com; 6Biological Sciences & Engineering, Indian Institute of Technology Palakkad, Palakkad 678623, India

**Keywords:** anti-malarial drugs, malaria vaccine, *Plasmodium falciparum*, vaccine adjuvants

## Abstract

Malaria—a parasite vector-borne disease—is a global health problem, and *Plasmodium falciparum* has proven to be the deadliest among *Plasmodium* spp., which causes malaria in humans. Symptoms of the disease range from mild fever and shivering to hemolytic anemia and neurological dysfunctions. The spread of drug resistance and the absence of effective vaccines has made malaria disease an ever-emerging problem. Although progress has been made in understanding the host response to the parasite, various aspects of its biology in its mammalian host are still unclear. In this context, there is a pressing demand for the development of effective preventive and therapeutic strategies, including new drugs and novel adjuvanted vaccines that elicit protective immunity. The present article provides an overview of the current knowledge of anti-malarial immunity against *P. falciparum* and different options of vaccine candidates in development. A special emphasis has been made on the mechanism of action of clinically used vaccine adjuvants.

## 1. Introduction

Malaria, caused by apicomplexan *Plasmodium* spp., remains one of the world’s most threatening diseases of humans and other animals with high morbidity and mortality rates [1]. The recent findings of the World Health Organization (WHO) recorded 228 million cases and 405,000 deaths globally in 2018 [2]. Although eight species of *Plasmodium* can infect humans, most malarial cases are due to *P. falciparum* or *P. vivax,* but deaths are mostly due to *falciparum* malaria. The African countries carry the highest share of the global malaria burden (~90% as per 2017 WHO reports) [3]. In Asian countries, e.g., India, which accounts for 4% of the global burden, malaria is still a serious health threat [4]. As an answer, the Special Programme for Research and Training in Tropical Diseases (TDR; co-sponsored by the WHO), US National Institutes of Health, UK department for international development, Bill & Melinda Gates Foundation, and other organizations have increased funding for research and development and other control measures (including vaccination) to minimize the malaria cases [2,5]. WHO has also put more effort into implementing essential malaria commodities, such as rapid diagnostic tests, insecticide-treated mosquito nets, vector control, artemisinin-based combination therapy, and in defeating insecticide resistance of malaria vectors. These programs were highly successful and did reduce significantly the numbers of malaria-related deaths [4,6,7]. 

The malaria parasite has a complex life cycle (Figure 1). The Ronald Ross and Giovanni Battista Grassi legacy laid the cornerstone for understanding the malaria parasite life cycle (Figure 1) [8]. There are pre-erythrocytic and erythrocytic stages in humans (host) and the sexual life cycle in the mosquito vector. (a) Pre-erythrocytic stage. When the infected mosquito bites the host for blood meal, it injects sporozoite mostly into the dermis or sometimes directly into the blood vessels [9]. Once sporozoite reaches the liver, it invades and infects the hepatocytes. Inside the hepatocytes, sporozoites multiply extensively, generating 10–30,000 merozoites [10]. (b) Erythrocytic stage. Merozoites enter the red blood cells (RBCs) and mature into trophozoite and schizont. Each schizont generates 6–12 merozoites, which are released into the circulation (remain one minute to the next infection) and infect more RBCs [11,12]. During the process, a small number of parasites develop into male and female gametocytes. The formed gametocytes mature first in the bone marrow (I–IV stages) and then spleen (V stage) [13]. (c) Mosquito/sexual stage. In every progressing cycle, a small number of parasites deviates from asexual reproduction and develops into male and female gametocytes, which will enter the stomach of the *Anopheles* mosquito during the blood meal. The male and female gametocytes develop into flagellated microgametes (eight) and a single macrogamete respectively within the mosquito’s midgut. A zygote resulting from the fusion of a macrogamete and microgamete develops into ookinete via meiosis and penetrates the mosquito’s gut wall to form oocysts and produce sporozoites. These sporozoites travel to salivary glands after the rupture of oocysts and will infect a new host during a mosquito bite (Figure 1) [12,14].

*P. falciparum* transmission can be managed through vector control approaches, such as spraying insecticides and/or using chemically-treated mosquito nets, and through the use of antimalarial drugs for prophylaxis and radical cure [15]. Early diagnosis, use of bed nets, and timely treatment with anti-malarial compounds are the global strategies for controlling *P. falciparum*. However, the parasite has developed resistance to all anti-malarial drugs which are in present use [16]. In the last century, chloroquine was an essential tool to eradicate malaria in many countries [17]. However, in the 1970s resistance to chloroquine monotherapy emerged, and nowadays, chloroquine-resistant *P. falciparum* parasites are present worldwide. Other drugs such as pyrimethamine, mefloquine, or artemisinin derivatives were also developed and replaced chloroquine progressively. Because resistance to these drugs also developed, combination therapies were set to treat *P. falciparum* infections [18]. Artemisinin-based combination therapies have been essential to decrease the death toll. However, resistance to artemisinin and its partner drugs has become a severe concern in the last decade [19,20]. Thus, there is a need for novel drugs and vaccines, particularly for controlling and possibly eliminating *P. falciparum* malaria. The focus of this article is to review the current status of malaria vaccines research, with an emphasis of the use of adjuvants and their mechanisms of action. 

## 2. Immunity against *P. falciparum*

Knowledge on the interaction between the parasite and host immune system and how clinical immunity is acquired against malarial infection is critical for the development of effective vaccines, anti-malarial drugs, and immunomodulators. 

### 2.1. Dendritic Cells and the Initiation of the Immune Response

Among the innate immune cells, professional antigen-presenting cells (APCs), such as dendritic cells (DCs), have a prominent role in antigen presentation [21]. Although each innate immune cell has a role in homeostasis and immunity, DCs’ role in vaccine and vaccine adjuvant development has been extensively studied [22,23]. Considering the central role of DCs in the activation and polarization of naïve T cells [24] and shaping the immune response to vaccine adjuvants (Figure 2), we focused our discussion on DCs [13] in the innate immune system. 

DCs appear to interact with the parasite at various anatomical locations of the host such as skin, liver, blood, and spleen. DCs recognize pathogens through specific receptors called pattern-recognition receptors (PRRs), such as toll-like receptors (TLRs), nucleotide-binding oligomerization domain (NOD)-like receptors (NLRs), retinoic acid-inducible gene-I (RIG-I)-like receptors (RLRs), and others. PRRs recognize highly conserved structures of microorganisms, including parasites, via pathogen-associated molecular patterns (PAMPs). PAMPs are processed by the DCs and present the unique sequences of PAMPs to the T helper (Th) cells, which seeds the development of immunological memory [22]. Although a few PAMPs (e.g., *Plasmodium* multidomain scavenger receptor-like protein [PxSR], glycosylphosphatidylinositol [GPI], hemozoin, RNA, plasmodial DNA, CpG motifs, and others) (Figure 2) [25,29,30,31] have been identified in *Plasmodium* spp., their interaction with DCs and the related immune defense are still not fully elucidated [32]. Interestingly, PRRs (particularly TLRs) have an essential role in both immune defense and pathogenicity (Figure 2) [33]. In addition, DCs have differential functional capabilities (tolerogenic versus inflammatory/activated) based on the tissues they reside in and signals they receive from the pathogen and microenvironment [34,35]. 

The malaria parasite in the mammalian host interacts with different tissues (the skin dermis, the liver, blood capillaries, and vessels) and can be either extra- or intracellular. This dictates which types of DCs it will interact with (for an in depth-review, see [35,36]).

In the dermis, the extracellular sporozoites are mostly phagocytosed by monocyte-derived macrophages (MDM) rather than by dermal DCs and monocyte-derived DCs. After sporozoites uptake, MDM display a regulatory phenotype, which negatively affects the antigen-presenting capacity of dermal APCs [37]. Nevertheless, some dermal DCs, after their interaction with sporozoites, could reach draining lymph nodes to prime CD4^+^ or CD8^+^ T cells [38,39]. Of all the sporozoites entering the blood circulation in their mammalian hosts, only a fraction can invade liver cells since most of them are retained in the spleen [40]. They can be phagocytosed by splenic DCs and presented to T and B cells. For the intracellular liver parasites, it is not clear how DCs acquire newly synthetized liver-stage antigens. It was shown that monocyte-derived DCs are responsible for CD8^+^ T cell priming [41]. However, the way they acquire antigen remains unknown. It has been proposed that DCs can either phagocyte apoptotic hepatocytes resulting from an aborted infection [38] or they can acquire antigens from infected hepatocytes [42]. It has long been known that intradermal immunization is less effective than the intravenous route when animals and humans have been injected with whole sporozoites [43,44]. This suggests that the priming of DCs in the spleen or liver is more potent than the skin’s priming. Since it is the natural route of infection, malaria parasites have evolved immune escape mechanisms at the skin. Identifying the adjuvants that can antagonize these mechanisms may help in the development of whole sporozoite skin immunization protocols. Of note, liver-resident DCs, which produce high amounts of IL-10, are more tolerogenic than other DC subsets. Cracking this tolerance, possibly through the use of proper adjuvants, represents one of the rational approaches to target liver-stage malaria infection [13,45,46]. 

During the erythrocytic stage, the more likely site of priming of the immune response is the spleen [47]. During its blood-stage development, the parasites remodel the infected erythrocytes, leading to changes in deformability and rheological properties of the cells [48]. The infected red blood cells (iRBC), which cannot deform, go through small capillaries and accumulate in the spleen. These iRBCs are then recognized and destroyed by phagocytes [49]. *P. falciparum* parasites have evolved to avoid splenic clearance by expressing neoantigens at the surface of iRBC. These antigens, such as *P. falciparum* erythrocyte membrane protein 1 (PfEMP1), allow parasites to sequester in deep tissues by binding to a variety of receptors expressed on the surface of capillary endothelial cells [50]. Splenic DCs or blood DCs can capture iRBC through CD36 and present to naïve T cells [51]. However, iRBC binding to CD36 and CD54 on immature DCs limits the DC maturation [52]. In vitro experiments in fact have shown that infected erythrocytes reduce the expression of co-stimulatory molecules on DCs [52] and affect the antigen processing capacity of these professional APCs [52]. It was also found that malaria infection alters the expression of TLRs on DC subsets [53]. In one cross-sectional study, circulating DCs from acutely infected Papuan adults had DCs, which did not mature well in vitro after stimulation and were more prone to apoptosis. This immune dysregulation of DCs is associated with an increased level of IL-10. Antagonizing the IL-10 actions restored DC functions [54]. By decreasing DC maturation efficiency, the parasite prevents the establishment of a potent immune response and facilitates immune evasion. Nevertheless, blood and splenic DCs are effective in inducing inflammatory signals against *Plasmodium* infection. New vaccine platforms are harnessing DCs for the development of novel malaria vaccines [55]. In addition, any immunopotentiating substances such as adjuvants, which can enhance the antigen presentation capacity of circulating or tissue-specific DCs to induce a protective immune response, are needed. 

Majority of the myeloid cells, such as monocytes, macrophages, NK cells, eosinophils, neutrophils, and DCs, participate in the antibody-dependent cell-mediated cytotoxicity (ADCC) functions. Previous studies have given a clue that passive transfer of immune sera has the capacity to activate ADCC to protect the host against malaria [56]. In vitro studies on RBCs, from infected and non-infected malaria, have confirmed that NK cell-mediated ADCC is essential for acquired immunity [57]. In addition, upon parasite exposure, NK cells are accompanied with an early capacity to produce copious amounts of cytokines (e.g., IFN-γ). Data from an RBC-supplemented, immune cell-optimized humanized mouse model have revealed that NK cells interact with *P. falciparum*-infected erythrocytes through lymphocyte-associated antigen 1 and kill the infected cells [58,59].

A specialized type of NK cells called adaptive NK cells, which express CD56^dim^ without the expression of promyelocytic leukemia zinc finger (PLZF) transcription factor and Fc receptor γ-chain (FcRγ) have been recently described. Studies conducted by Geoffrey et al., in children and young adults, have predicted that presence of adaptive NK cells is associated with protection against *P. falciparum*-induced malaria infection [60]. As NK cells are heterogeneous, further work is necessary to dissect the role of various subsets of NK cells in the protection against malaria. Nevertheless, based on the above discussion, ADCC function might play an essential role in blood-stage malaria. 

### 2.2. Adaptive Immunity

#### 2.2.1. T Cell-Mediated Immunity

T cells, such as CD4^+^ (T helper, T follicular helper (Tfh) cells, cytotoxic CD4^+^ T cells), cytotoxic CD8^+^, and non-classical T cells such as mucosal-associated invariant T (MAIT) cells and γδ T cells have been shown to have a role in antimalarial immunity (for a comprehensive review, see [61]). 

#### 2.2.2. T Cell-Mediated Immunity against Liver-Stage Malaria Parasites

Most of the knowledge on T cell immunity against the malaria liver stage has been obtained in rodent malaria models of protection using whole parasite immunization either with irradiated sporozoites or live sporozoite under drug prophylaxis [62,63,64,65] (Table 1). In these models, it has been shown that CD8^+^ and CD4^+^ T cells recognize peptide-derived antigens, which are presented by MHC class I and Class II on the surface of infected hepatocytes [66,67,68,69]. T cells can eliminate infected parasites via different mechanisms [70]. They can produce IFN-γ, which induces nitric oxide synthase to produce nitric oxide, a toxic molecule, in response to liver parasites [71,72,73]. They can also eliminate infected hepatocytes through contact-dependent perforin-dependent mechanisms [74]. IL-4-secreting CD4^+^ T cells have been shown to be critical for the induction of CD8^+^ T-cell response to liver-stage malaria [41,75,76]. Liver-resident CD8^+^ T cells (CXCR6^+^CD69^+^) induced by vaccination with whole parasites or a subunit formulation of the *Plasmodium* ribosomal protein RPL6 are essential to confer protection in mice [77,78]. Experimental vaccination experiments in models using irradiated sporozoites have also suggested that protection is dependent on the presence of liver-resident CD8^+^ T cells [79]. In human vaccine experiments, CD8^+^ and CD4^+^ T cell responses were induced after *P. falciparum* sporozoite immunization [80]. Vaccine-induced protection was correlated with CD4^+^ T cell responses [80]. The route of infection is critical for optimum T cell-mediated immunity against liver parasite. In fact, intravenous but not the intradermal injection of irradiated sporozoites induced a full sterilizing immune response [79], possibly due to the induction of resident CD8^+^ T cells [44,79]. Therefore, vaccines, delivery systems, or adjuvants which that can induce liver-resident T cells and increase antigen processing and presentation by MHC molecules at the surface of infected hepatocytes are highly desirable. This has led to strategies using viral vectors and heterologous prime-boost approaches [80,81,82]. Recently, a novel and promising approach called ‘prime and target’ has been shown to increase the frequency of tissue-resident memory CD8^+^ T cells in the liver in a mouse model using intravenous injections of viral vectors or nanoparticles [81]. Other cell types, such as γδ or MAIT cells, may also have a significant role in mediating the *P. falciparum* sporozoite vaccine-induced protection as suggested by the studies done with adults in the United States and Mali [80,83,84]. However, it is yet to be determined how these cells are induced during vaccination and what the best adjuvant will be to activate them to act synergistically with classical T cells for an effective protective immunity. 

#### 2.2.3. T Cell-Mediated Immunity against Asexual Blood-Stage Malaria Parasites

T cells are essential for the control and resolution of blood-stage infections. Experiments in rodent malaria models have shown that a CD4^+^ Th1 response is needed to control the first phase of infection by limiting the parasitemia [85,86,87]. Resolution of the infection is mediated by Th2 and Tfh CD4^+^ cells, which provide help for efficient protective antibody production [85,86,87]. γδ T and NK cells have been shown to be able to kill the intraerythrocytic parasites in vitro, but their exact role in protection remains a debated question [83,84,88,89].

Until recently, CD8^+^ T cells were thought to have no role in the protection against blood-stage malaria. However, experiments in rodent models using mice deficient in programme death (PD)-1, a marker of T cell activation and exhaustion, have demonstrated an effector role for this T cell subset [90,91]. Because normocytes, the mature erythrocytes, lack MHC molecules, T cells cannot directly eliminate the parasite inside RBC. Malaria-specific T cells secrete IFN-γ, which activates monocytes/macrophages for phagocytosis and elimination of iRBC [90,92]. A handful of studies in humans have shown that CD8^+^ T cell activation occurs during falciparum infection. In a recent study in Ghanaian children, expansion of a subset of CD8^+^ T cells expressing granzyme B correlated with an increase in parasitemia and was more pronounced in patients with severe disease [93]. Thus, it is likely that CD8^+^ T cells may have a more pathogenic effect than protective role during falciparum infection. This is supported by immunohistopathology investigations on the brain vasculature of 31 children who died from human cerebral malaria. A substantial number of CD3^+^ CD8^+^ T cells was found adjacent to endothelial cells [94]. In a mouse model of cerebral malaria, pathogenic CD8^+^ T cells are responsible for the neurological symptoms and ensuing lethality induced by infection with *P. berghei* ANKA [95,96]. These cells recognize malaria antigens presented by endothelial cells and release granzyme B after engagement with the MHC molecules. Thus, for blood-stage vaccine development, induction of a strong immune response may not be desirable since it may induce unwanted pathology. Nevertheless, there is a need for well-calibrated T cell and well-targeted B cell responses.

As already mentioned above, malaria infection frequently induces immunosuppression through diverse mechanisms [97]. One of the active T cell suppression pathways involves the induction of regulatory T cells (Tregs). Tregs expressing specific markers such as cytotoxic-T-lymphocyte-associated protein-4 (CTLA-4), lymphocyte-activation gene 3 (LAG-3), programmed cell death-1 (PD-1), and others function as immunosuppressive cells. Both in vitro studies in peripheral blood mononuclear cells (PBMCs) and longitudinal studies in humans have shown that the frequency of Tregs in the patients is negatively associated with the parasite load and/or disease severity [96,98] (extensively reviewed elsewhere [61]). Studies in humans and mice have shown that Tregs via CTLA-4 inhibit the development of a protective anti-blood-stage immunity [99]. Furthermore, clinical studies on malaria-infection/vaccination showed an increased Treg population in the acute and uncomplicated infection and during the convalescence phase [100,101,102]. In contrast, a decrease in the Treg population was also observed in children under chronic malaria exposure [103,104,105]. Thus, knowledge of the factors influencing Treg functions, such as age, level of exposure (high or low transmission intensity), and others, is needed to precisely define their role in the pathogenesis of malaria. Approaches using adjuvants to prevent induction or counteract Treg activity in individuals residing in endemic regions might help in the development of an effective anti-blood-stage vaccine. 

#### 2.2.4. Antibody-Mediated Immunity

In malaria infection, antibodies mediate protective immune responses via different mechanisms, such as inhibition of parasite motility, invasion, egress, adhesion and hepatocyte traversal ability, promotion of antibody-dependent complement-mediated sporozoite/merozoite lysis, phagocytosis, antibody-dependent cellular cytotoxicity, transmission-blocking activity, and others [80,106,107] (reviewed in [108]). Interestingly, the generation of natural immunity against *P. falciparum* is ethnicity dependent. The majority of non-immune Western travelers, if not all, require a single infection to induce invasion inhibitory antibodies against *P. falciparum* rather than endemic residents, who require two infections [109,110]. Similarly, protection against malaria in infants born in the endemic regions correlate with the pre-existing antibodies (i.e., maternal antibodies) [110]. In addition, children (5 years) who are repeatedly exposed to the natural infection are more resistant to the severe clinical symptoms of malaria [110,111]. Development of *P. falciparum* antigen-specific B cells to secrete monoclonal antibodies and passive transfer of immunoglobulins have been attractive strategies in malaria research [80,112].

Many malaria candidate vaccines have been designed to induce an effective antibody response. Antibodies against sporozoites or against neo-antigens expressed on the surface of infected hepatocytes mediate protection by preventing or limiting pre-erythrocytic-stage infection and development. Anti-sporozoite antibodies have been shown to inhibit sporozoite motility in the dermis and liver [113], destroy sporozoites in the skin [114], facilitate opsonization and phagocytosis by monocytes or macrophages in the spleen or the liver [114,115], inhibit sporozoite invasion into hepatocytes [116], and inhibit sporozoite development inside the hepatocytes [116]. The type of functional antibodies produced during the infection illuminates the type of protection they offer. For example, IgG1 and IgG3 exhibit potent capacity to activate complement, FcɣR signaling and opsonization compared to IgG2 and IgG4. The former function discriminates the IgG subclasses into cytophilic (IgG1, IgG3) and non-cytophilic (IgG2, IgG4) antibodies [117,118,119]. The above-mentioned antibody functional properties have been confirmed against the majority of *Plasmodium* spp., including *P. falciparum* in different life-cycle stages [108]. Although the role of IgG and its subclasses is significant in host to symptomatic malaria infection, the role of IgM must not be ignored. Particularly, the protective role of IgM against the blood-stage parasite of *P. falciparum* has been established [120,121,122]. 

Antibodies against parasite neo-antigens, such as heat shock protein, expressed on the surface of infected hepatocytes induce liver parasite killing through an antibody-dependent cell-mediated mechanism, which is likely to involve Kupffer cells or NK cells [123]. Different in vitro assays have been developed to measure antibody functionality, but it is not yet clear which of the immune mechanisms described above are essential or associated with the protection. It might be that, in addition to the high level of neutralizing antibodies, the quality (i.e., avidity, affinity, isotype) of the antibodies is also important. This knowledge is of paramount importance to design better immunogen and use adequate adjuvant. 

Antibody-mediated immunity against the blood stage is more complex than in the pre-erythrocytic stage. Anti-merozoite antibodies can (i) prevent merozoites from invading RBC [124,125,126] alone or in conjunction with complement factors [118], (ii) prevent merozoite egress from RBC, (iii) agglutinate free merozoites, (iv) facilitate phagocytosis of merozoites, and (v) promote clearance of iRBC by phagocytic cells through a mechanism called antibody-cell-dependent inhibition (ACDI) [127]. In ACDI, anti-merozoite IgG1 or IgG3 antibodies bind to merozoites, and the immune complexes promote phagocytes, such as monocytes/macrophages or neutrophils, to release cytokines (e.g., TNF-α). This cytokine then stimulates the phagocytes to produce mediators, which leads to the killing of intra-erythrocytic parasites [128,129]. As mentioned above *P. falciparum* parasites express antigens on the surface of iRBC. These antigens are mainly encoded by multigene families, such as the *var* [130], *stevor* [131], and *rifin* gene families [132]. These antigens are involved in cytoadherence to endothelial cells and in other adhesive phenomena, such as rosetting (the binding of an iRBC to non-infected RBC) and agglutination (the binding to iRBC through bridging by platelets) (for a review, see [50]). The cytoadherence ability of the malaria parasites is key to many of the pathologies induced by *P. falciparum* infection. Antibodies targeting the surface antigens may prevent cytoadherence and promote iRBC phagocytosis or iRBC agglutination [133,134]. 

Antibodies targeting parasite toxins could also protect from *P. falciparum*-induced disease. During the blood stage of the infection, diverse parasite toxins are released at the time of iRBC rupture. These toxins include hemozoin, a by-product of heme degradation by the parasite [135], GPI moieties, which are present in many merozoite proteins [136], a TatD-like DNase [137], and a tyrosine-t RNA synthase [138]. Protection from disease by anti-toxin antibodies has been achieved experimentally using synthetic glycans mimicking GPI [139]. 

Gametocytes, the sexual forms of *P. falciparum* parasites, are also targets for antibody-mediated immunity [140]. Gametocytes express a range of antigens, which are targeted by antibodies [141]. The latter facilitates the complement-mediated killing of the gametocytes [142]. In the mosquito after feeding, anti-sexual form antibodies can prevent fusion of gametes [143], induce complement-killing of gametes or ookinetes [144], and prevent ookinete motility, penetration of the midgut wall, and formation of oocyst [145]. 

Vaccination/infection determines the dominant IgG subtypes induced, which in turn are useful to predict the type of immune responses acquired. Among the IgG subtypes, opsonizing cytophilic antibodies have gained much attention for malaria vaccine development [146,147,148]. These antibodies recruit FcɣR-containing immune cells (particularly macrophages and basophils) via their Fc domain and activate the parasite elimination procedures. Cytophilic antibodies both naturally acquired [149,150,151,152] and vaccine-induced (IgG1 and IgG3 responses against blood-stage antigens) [107] have shown more protective responses in diverse studies. 

During the vaccine design, it important to consider both naturally acquired and vaccine-induced immunity. The development of naturally acquired immunity depends on many factors such as region, age group, number of exposures (symptomatic or asymptomatic), number of targeted antigens, and others [149,150]. The vaccine-induced immunity depends on the type of antigens targeted and the quality (type) and quantity of antibodies produced [153]. Although both have similar specificities, the duration and diversity of the immune response is more for naturally acquired immunity over vaccine-induced immunity [154]. Evidence from different studies suggests that naturally acquired immunity, unlike vaccine-induced immunity, reduces the risk of infection irrespective of age [155,156]. Moreover, children living in perennial transmission areas induce more protective natural antibodies (e.g., anti-erythrocyte binding antigen 175RIII–V) than in seasonal transmission areas [149]. However, development of naturally acquired immunity is the best bet in the host. Circumstantially, newborns and infants (<6 months of age) are conferred protection against malaria infection due to the existence of maternal antibodies (extensively reviewed elsewhere [157]). 

Antibodies’ breadth (antibodies raised against number of antigens), magnitude (quality), and quantity (affinity and avidity) decide the protection efficiency against clinical malaria. However, it is difficult to estimate the quantity of antibodies that are required for protection, and it purely relies on the antigen specificity. A study conducted in Kenyan children (*n* = 119), for six months, confirmed that children who possess a breadth of naturally acquired antibodies against blood-stage antigens are inversely correlated with the risk of malaria [158]. With regard to the vaccine development, whole-sporozoite vaccines are ideal to obtain a breadth of antibodies against the parasite. Like natural exposure, whole-sporozoite vaccines increase the magnitude of antibodies due to the varying antigenic content [80]. 

Interestingly, it has been observed that antibody breadth and magnitude increase with age (4 years < 14 years < adults) [159]. Though not always, increased risk of infection has also been observed in the presence of antigen-specific antibodies (e.g., anti-AMA1) [160,161]. However, additional studies are required to confirm the breadth, magnitude, affinity, and avidity of the protective antibodies in both asymptomatic and symptomatic infections [162,163]. 

As mentioned above, passive transfer of neutralizing monoclonal antibodies is an alternative therapeutic strategy, which gives short-term protection. In addition, epitope-specific monoclonal antibodies have been generated against different stages of *P. falciparum* and evaluated in several models (reviewed, recently) [112]. At present, subunit, DNA, and viral vector vaccines are capable of inducing high antibody titers, though the potency and breadth are not sufficient to give complete protection [164]. To achieve more germinal center reaction and memory B cell generation, the addition of vaccine adjuvants and adjuvant delivery systems are essential (discussed in the later section). Finally, natural or vaccine-induced high antibody titers are needed to prevent the hepatic entry of the parasite and erythrocyte infection. 

## 3. Vaccines

Vaccines are one of the most efficient prophylactic treatments for many diseases, ranging from smallpox to the recently emerged severe acute respiratory syndrome coronavirus 2 (SARS-CoV-2, which causes coronavirus disease 2019 (COVID-19)). With the ever-increasing knowledge of immunity, vaccine strategies have focused on identifying the target antigens and inducing a protective immune response with low to no side effects. Due to the complex antigenicity and immune attacking mechanisms of *P. falciparum*, the development of vaccine for this parasite has been and is still challenging (Appendix A) [165]. However, many traditional vaccine strategies do not answer the question of life-cycle complexity and polymorphism of parasite proteins, which are just two of the many reasons we do not yet have a highly effective malaria vaccine [80].

A variety of malaria vaccines have been developed and evaluated from the past decades, ranging from classic approaches such as whole inactivated parasites to subunit vaccines and new delivery systems. Although adjuvants enhance vaccines’ immune response, they are occasionally not efficacious in formulations with malarial antigens. Different recombinant malarial antigens have been produced (Table 1 and Table 2) in various expression systems (e.g., *E. coli*, *L. lactis*, *N. benthamiana*, *P. pastoris*, *S. cerevisiae*, and others) with a variety of delivery platforms (e.g., conjugates, fusion proteins, virus-like particles, virosomes, and others). Over the years, researchers have used various antigens for vaccine development against the different stages (sporozoite, merozoite, and sexual) of the parasite life cycle [166]. However, it has been challenging to identify the ideal target antigen, among the 5000 proteins encoded in the *P. falciparum* genome [167]. Genomic, transcriptomics, and proteomics approaches have helped to define protein expression patterns during the *P. falciparum* life cycle [168]. Among the proteins identified, the uniqueness of sporozoite, trophozoite, merozoite, and gametocyte proteins are categorized [168]. The different protein targets (mentioned in Table 1 and Table 2) are exploited in distinct ways for developing malaria vaccines: (i) pre-erythrocytic vaccines, (ii) erythrocytic vaccines, (iii) transmission-blocking vaccines (extensively reviewed elsewhere [80,110,169]). Several candidates have been tested; several immunogens are at preclinical stage, among which 30 immunogens are in clinical evaluation though only 2 have crossed Phase IIb trials or beyond (without adjuvants; https://clinicaltrials.gov/ accessed on 1 June 2020). 

In the past 100 years, many adjuvants have been evaluated preclinically against diverse antigens of malaria (Table 2). Notably, RTS’S/AS01 (Mosquirix™) (Appendix B) and PAMAVAC vaccines have successfully entered advanced clinical trials.

## 4. Vaccine Adjuvants

In general, most vaccine-based antigens are inefficient at mounting protective immune responses due to the dearth of immunogenicity, even though they proved to be perfect target antigens. Nevertheless, an additional component, named adjuvant (*adjuvare* (Latin): To help) has proven to be critical in boosting the antigens’ immunogenicity. Several reviews have covered the qualities of most vaccine adjuvants and their mechanism of action [170,171,172,173,174,175]. Like immune response in infections, adjuvants induce an immune response by triggering innate and adaptive immune responses. The type of immune response induced by the adjuvant is critical for the choice of vaccines. For example, adjuvants inducing Th1 response are preferable in vaccines against intracellular pathogens, and adjuvants inducing Th2 responses are preferable in vaccines against extracellular pathogens [174,176]. In both cases, Tfh cell responses are critical in inducing prolonged antibody responses [177]. Tfh cells are distinguished from the other T helper cells via expression of a CXCR5 receptor and a transcription factor B cell lymphoma 6 (Bcl-6). During the infection or vaccination (with or without adjuvants) [178], Tfh cells migrate to the secondary lymphoid organs and provide help to the germinal center B cells to induce high-affinity memory B cells and long-lived plasma cells [177,179]. The role of Tfh cells in humoral immunity has become more motivating in the design of subunit vaccines [180]. 

Protective immunity against *P. falciparum* requires neutralizing antibodies [181,182] and optimal Th1-mediated immunity [183]. For enhancing, modulating, and prolonging the specified immune response against any *P. falciparum* vaccine candidate, vaccine adjuvants that facilitate these responses are needed. These adjuvants will reduce the antigen concentration and frequency of immunization to attain protective efficacy, thereby making vaccines more cost-effective. Though adjuvants face many challenges (physicochemical interactions, stability, pairing with partner antigen, and others), recent technologies in vaccine adjutants have come to the stage of increasing immune responses via immune synergy (use of multiple adjuvants) [172]. Reasonably, the immune synergy strategy has not been used extensively in the vaccines composed of parasitic antigens [184]. 

Different types of adjuvants have been categorized, such as TLRs-, nucleotide-binding oligomerization domain-like receptors (NLRs)-, C-type lectin receptors (CLRs)-based, and some other non-specific PRRs agonists, NLR family pyrin domain containing 3 (NLRP3) activators, formulations including liposomes, Adjuvant System (AS), nano/micro particles-based adjuvants, immune-complexes, and others. Recent years have seen the approval of several new-generation adjuvants containing vaccines for human use [185]. Inflammation (either specific receptor activation or non-specific activation) is the primary response to a majority of the adjuvants leading to antigen-specific cellular immunity [185]. Among the long list, formulation vaccines have gained much attention with respect to their efficacy, such as AS01, AS02, AS03, AS04, MF59, and others. The majority of these systems have more than one immune-potentiator in them. The use of these formulations has induced a protective immune response against antigens and has also increased the immune responsiveness in an elderly population, e.g., influenza vaccine (MF59 or AS03) and herpes zoster vaccine (AS01) [185]. Although many adjuvants, including alum (a century-old approved adjuvant), are used in clinical trials, the lack of a clear understanding of mechanisms of action against malaria has impaired the development of efficient immunogen/adjuvant formulation [186]. Viral vector-based vaccines have shown better immunogenicity than adjuvanted protein vaccines [187]. Herein, we discuss the importance of current adjuvants, which are being employed in developing malaria vaccines at the clinical stage, focusing on their outputs like immunological profile and mechanism of action of successful adjuvants.

## 5. Adjuvants under Clinical Evaluation

### 5.1. Alum

Alum (common name for aluminum potassium sulfate) is a 90-year-old gold standard vaccine adjuvant that produces primarily humoral immune responses [188,189]. A recent review detailed the physico-chemical and biological properties of alum adjuvants [188]. Mostly, two types of aluminum adjuvants (aluminum hydroxide (Al(OH)_3_) and aluminum phosphate (AlPO_4_)) are used in the clinically approved vaccines and these are the most widely applied adjuvants for evaluating the malaria vaccines [190]. The merits of aluminum adjuvants include safety, amplification of antibody responses, and a comparatively easy process to produce on a large scale. It also has certain drawbacks like the inability to induce cell-mediated immune responses (Th1 and cytotoxic T lymphocytes (CTL)), which are essential for *P. falciparum* [190]. Since alum induces Th2 biased cellular responses, it was administered concomitantly with other adjuvants (alum+CpG motifs (ODN) [191] alum+glucopyranosyl lipid adjuvant (GLA) [192]) or adjuvant formulations in the development of malaria vaccines. In preclinical studies, often, the antigen is mixed with alum before injection in mice [193,194]. Unlike other adjuvants, alum is the most successful single vial- or prefilled syringe-based vaccine adjuvant. Though intense research has been done on the alum, its precise mechanism of action, however, is still elusive. Interestingly, studies in mice showed that alum forms a depot at the injection site and enters the innate cells via phagocytosis. It activates the NLRP3 inflammasome via phagolysosomes followed by the activation of caspase-1 and release of IL-1β. The known mechanism of action of alum was reviewed elsewhere [189]. Nevertheless, the proposed mechanism is not observed either in humans or in monkeys [195,196]. Thus, there should be another mechanism of action of alum in humans, which is yet to be identified [185]. The vaccination of multistage antigen SpF66 combined with alum resulted in the development of short-lived antibodies and minimal cellular response with no or little protection. Further, the evaluation of pre-erythrocytic protein Pfcs102 with alum was not successful in enhancing an antigen-specific antibody response [190,197]. Blood-stage vaccines (AMA1[PfAMA-1-FVO_25–545_], AMA1-C1, GLURP_85–213_, and MSP1-C1_42_) adjuvanted with alum induced modest antibody levels, a poor cellular response, and no protection in field clinical trials [190]. In a randomized, double-blind, placebo-controlled study, one vaccine (PRIMVAC) adjuvanted either with alhydrogel (aluminum hydroxide wet gel suspension; regularly 2% Al(OH)_3_ is used) or GLA-SE had a significant immunogenic response and VAR2CSA-specific antibodies with the desired inhibitory properties [198]. 

The transmission-blocking Pf25, a sexual-stage vaccine, has been clinically evaluated in combination with alum. It induced local reactogenicity and limited immunogenicity [199,200]. Altogether, it is clear that alum alone is not sufficient to induce effective responses to malaria vaccines [201]. 

### 5.2. Vaccine Delivery Systems/Formulations

Although effective or moderately effective protein vaccine candidates have been developed for malaria, they have poor immunogenicity. Targeted delivery of subunit vaccines via systems possessing adjuvant properties is of paramount importance and a very classical approach. It ensures their effective delivery as well as the ability to increase protective immunity [166]. However, like alum, the mechanism of adjuvanticity of emulsion-type delivery systems remains to be identified [185]. 

#### 5.2.1. Liposomes 

They are synthetic phospholipid models (nm to µm size), which carry antigens encapsulated into the aqueous core (hydrophilic molecules), adsorbed to the surface (lipophilic molecules), or integrated into the lipid layers (amphiphilic molecules) [202]. The adjuvanticity of liposome is mainly reliant on size, charge, preparation method, and number of lipid layers [203]. Cationic adjuvant formulations made of dimethyl dioctadecyl ammonium combined with the stabilizer glycolipid trehalosedibehenate (TDB) were found to be immunogenic liposome forms [204]. The disadvantages of liposomes are stability, manufacturing process, high costs, pain at the injection site, and the addition of immunostimulatory molecules like AS01 [190]. Because of the instability of liposomal vaccine delivery systems in a single vial liquid format, the formulation should be admixed with the vaccine before administration, e.g., Mosquirix^®^ vaccine [201]. 

#### 5.2.2. AS01

AS01 is a liposome-based vaccine adjuvant system containing two immunostimulants: 3-O-desacyl-4′-monophosphoryl lipid A (MPL) and the saponin QS-21. It has been extensively studied for vaccines against various infectious diseases, including *catarrhalis*, influenza, tuberculosis, HIV, and others [205]. Interestingly, unlike other vaccine adjuvant systems, AS01 has a recognized mechanism of action. It activates the resident macrophages, followed by draining to the lymph nodes, where it activates the NK cells to release IFN-γ. The released IFN-γ activates DCs, which induce an antigen-specific Th1 response (i.e., IL-2, IFN-γ, and TNF-α) [206,207,208]. The above-mentioned mechanism of action is due to the synergistic effect of two immunostimulants, i.e., MPL (monophosphoryl lipid A) and QS-21, present in AS01 (Figure 3). Both the molecules induce Th1-mediated immune responses [209]. In addition, in delayed fractional doses AS01/RTS, S has been shown to induce IL-21-secreting antigen-specific peripheral Tfh cells with protective B cell responses [210]. However, individual components have their specific effects (see the relevant text). QS-21 delivered via liposomes protects the cells from QS-21-induced cell death [211]. This might be the reason that QS-21-containing AS01 has exhibited more improved efficacy than other formulations. In addition, malaria vaccine clinical trials have used different forms of AS01, i.e., AS01B full dose of immunostimulants for adults and AS01E half dose of immunostimulants for children [185,212] (Table 2).

**Note: Glucopyranosyl Lipid Adjuvant (GLA) is a synthetic hexaacylated lipid A derivative**. Monophosphoryl lipid A (MPL^®^) is primarily produced from *Salmonella minnesota* R595 by eliminating phosphate from the reducing-end glucosamine, core carbohydrate group, and the acyl chain from the 39-position of the disaccharide backbone [213].

#### 5.2.3. Emulsions

Emulsions are biphasic liquid dosage forms composed of three main components, oil phase, aqueous phase, and emulsifying agents (surfactants), and classified under conventional, widely used vaccine delivery systems. They are water-in-oil (W/O), oil-in-water (O/W), and multiple emulsions (water-in-oil-in-water (W/O/W) and oil-in-water-in-oil (O/W/O)). Among the emulsion types, O/W type emulsions are highly preferred because of less reactogenicity and sustained release. For example, oleic acid O/W emulsion with monophosphoryl lipid A (MPLA) and squalene-based O/W emulsion with murabutide induce a significant cellular as well as high avidity humoral immune response in mouse models [214,215]. A potent O/W emulsion-based immunological adjuvant, MF59, which is composed of squalene droplets stabilized with Tween 80 and Span 85, elicited higher antibody titers with balanced antibody subclasses than those observed in alum studies. In contrast to conventional emulsions, MF59 is composed of metabolizable oils (squalene; component in cholesterol synthesis pathway), making MF59 much safer. Strikingly, although MF59 is more highly efficacious, stable, and potent than other vaccine adjuvants (alum and CpG) with ease of scale-up competence [185,216], it did not offer reasonable results in malaria vaccines; hence, there was no further progress with MF59 in developing malaria vaccine [217,218,219,220]. Irrespective of antigens, O/W emulsions elicit Th1/Th2 responses rather than Th1/Th17 responses [221]. Other emulsion-based adjuvants have been developed and have shown to be more efficient. 

#### 5.2.4. AS02

It is composed of two immunostimulants: QS-21 and MPL (3-deacylated monophosphoryl lipid A) in squalene O/W emulsion. Preliminary results in rodents and non-human primates have shown that AS02 is efficacious [222,223]. Further, early clinical trials of AS02 with different malarial antigens, such as PfCS102 (a *P. falciparum* circumsporozoite protein immunogen) and LSA-1 (liver-stage antigen) have shown an increased Th1 response [224]. Initial trials with RTS, S/AS02 formulation (AS02A and AS02D are varying doses of formulation for adults and infants, respectively) have found that the vaccine is safe, well-tolerated, and immunogenic in adults, infants, children, and semi-immune adults [225,226,227,228,229]. Interestingly, RTS, S/AS02 formulation exhibited effective cellular immunity and significant protection against sporozoite-challenge malaria infection when compared to other AS formulations [186,212,230]. However, further trials, with distinct dosage regimens have reported no or minimal protection [186,212,231]. In view of these outcomes, among the AS formulations for malaria vaccines, AS01 holds a superior efficacy and has been selected for further clinical trials [186,212,225]. 

#### 5.2.5. Montanides (ISA 51, ISA 720)

Montanide ISA 51 and ISA 720 are W/O type emulsions, which are composed of mineral oil and non-mineral oil, respectively, with mannide monooleate as an emulsifier. ISA 51 was approved for lung cancer due to its high CTL responses. Clinical studies on ISA 51 adjuvant with mosquito-stage antigen (Table 2) yielded unexpected systemic adverse events (erythema nodosum), which stopped further exploratory studies in malarial vaccines [232]. ISA 720 has been extensively studied in malaria vaccines because of its ability to induce high antibody production rather than T cell responses. In most of the malaria vaccines, ISA 720 has been shown to induce a strong immune response, but with associated problems such as pain and unacceptable reactogenicity at the site of injection and formulation instability [190,233]. All these findings further support that O/W emulsions are preferred in human vaccines. 

### 5.3. Immune Potentiators or Immunomodulators

Substances, such as particulate materials, complexes, delivery systems, and others, which modulate the immune responses either by targeting the innate immune system or skewing the Th1- and Th2-dependent antibody production, are referred to as immune potentiators. Many molecules derived from the classes of structure-guided vaccine adjuvants or immune potentiators, e.g., PamCysk4 analogues (TLR2 agonists), α-Glactosylceramide analogues (CD1d-iNKT linkers), saponins/glycolipids, and others, show immune-modulating potential [193,194,234,235,236,237,238,239]. In this section, we describe different potent immunomodulators, which are potent inducers of Th1 responses and have been used in malaria vaccines, either alone or in combinations.

#### 5.3.1. QS-21

QS-21 saponin is derived from a natural source, the bark of *Quillaja saponaria* Molina. Very recently, the mode of action of QS-21 was documented. After injection, it is drained to the lymph nodes and activates subcapsular macrophages (CD11b^+^CD169^+^), where activated macrophages recruit neutrophils and DCs. DCs undergo maturation and induce antigen-specific T cell (CD4^+^ and CD8^+^) and antibody responses. Interestingly, further studies confirmed that QS-21, like ISCOMATRIX and alum, induces the activation of inflammasome and the release of IL-1β. Inflammasome activation is mediated via the MyD88 pathway with the partial activation of high-mobility group protein B1-TLR4 (Figure 4) [240]. QS-21 (in AS01)-dependent DC activation is mediated through cathepsin B. Moreover, QS-21-mediated Syk kinase activation and lysosomal destabilization are essential for monocyte-derived DC activation and antigen-specific immune response (Figure 4) [241]. Nevertheless, a structural-activity relationship clearly suggests that triterpene aldehyde and fatty acyl side chains are the vital moieties for QS-21 action. Strikingly, deacylation reduces the induction of Th1-type responses [211].

Though it is effective in inducing immunity and widely used in many vaccines, including malaria, HIV-1, and cancer, QS-21 also possesses certain limitations, such as complex steps of manufacture/synthesis, hemolytic effects, biodiversity problems (due to the cutting down of trees), the dose to efficacy ratio, and others [174]. Considering all these limitations, novel structural mimics and alternative or simplified structures are under progress [239]. Due to the virtue of its immune response, many formulations are use QS-21 as a discrete component in the composition [185]. Interestingly, QS-21 containing nanopatch–skin delivery systems have been shown to be effective for influenza protein antigen. In this study, a nanopatch, which contains a 30-fold lower dose of QS-21, has exhibited an equal antigen-specific humoral response to QS-21 alone, given intramuscularly [242]. This clearly suggests that in the future, QS-21 with malarial antigen must be tested in nanopatches or a similar delivery system to spare the doses of both antigen and adjuvant.

#### 5.3.2. Other Saponin Based Adjuvants (Quil-A and ISCOMs)

*Quillaja saponaria* is the base for Quil-A and is the most widely used adjuvant from the saponin group. It exhibits lower toxic effects but enhanced adjuvanticity. Further, complex saponin adjuvant, immunostimulating complexes (ISCOMs; Matrix-M™) (spherical, open, and cage-like structures) formed by the cholesterol and phospholipids with the adjuvant (Quil-A or QS-21), trigger both cell-mediated and humoral-mediated immune responses. On the other hand, Quil-A is complicated on account of its chemical-related problems plus local reactogenicity. Although malaria vaccine candidates combined with ISCOMs have shown good results in preclinical investigations [243], very few clinical trials have been conducted with malarial vaccines composed of ISCOMs (NCT02905019), probably due to the pain at the injection site and other drawbacks such as flu-like symptoms, fever, and malaise with less reactogenicity [190,244,245].

#### 5.3.3. CpG ODN

Oligodeoxynucleotides (ODNs) adjuvants are single standard DNA molecules with varying degrees of unmethylated **CpG** motifs. Like most vaccine adjuvants, CpG ODN also enhances the innate immune cell (pDCs, monocytes, NK cells)-dependent adaptive immune responses (Th1 and B cells) against the co-administered antigens [246]. Considering the strong immunomodulatory properties, especially Th1 responses, CpG ODNs are widely used in various diseases such as infections, cancer, and allergies [246]. Based on their structural-activity relationship, CpG ODNs are divided into different types: class A (Type D), class B (Type K), and class C [174,246] (reviewed in [174]). Among the ODNs, ODN 2006 (7909) is widely used because of its immunostimulatory properties. It comes under the class B category, and it stimulates pDCs to secrete IFN-α via binding to the TLR9 receptor. The detailed mechanism of action of CpG ODNs has been reviewed in the literature [174,246]. Many preclinical studies, either alone or in combination, have documented the efficacy of CpG in *P. falciparum* infection [247]. This ODN has successfully entered the malaria vaccine clinical trials and has been used as an adjuvant for malarial antigens, such as AMA1, MSP1(42), and their different combinations (Table 2). In one of the clinical studies, naïve human volunteers who received AMA1-based blood-stage malaria vaccine along with CpG ODN, either alone or in combination with alhydrogel, exhibited enhanced and long-lasting antigen-specific antibody responses. In addition, antigen-specific antibody (IgG) showed enhanced growth inhibition of homologous *P. falciparum*, in vitro [248]. However, subsequent studies with semi-immune adults living in Mali did not give encouraging results [249]. The above results perhaps depend on the refractory/chemistry of CpG ODNs and should be taken into consideration for the future use of CpG ODN-containing vaccines in malaria endemic areas. One more blood-stage vaccine candidate, serine repeat antigen 5 (SERA5), has shown protective immunity in *Aotus* and squirrel monkeys. A recombinant antigen, SE36 (which lacks serine repeats of SERA5), was absorbed into alhydrogel and evaluated in clinical trials. Surprisingly, it did not induce high antigen-specific antibody titers in clinical trials [250]. Further, SE36/alhydrogel was tested in rodents and non-human primates (monkeys) with or without CpG ODNs. Among the CpG ODNs tested, K3 CpG ODN (K3 CpG ODN is a subclass of ODNs under class B (Type K)) has exhibited safe, efficacious, and immunogenic properties [251]. However, with CpG ODN, these positive preclinical data are still not translated into clinics [252].

## 6. Concluding Remarks

*P. falciparum* is an infectious agent of malaria, which kills millions of people around the globe. A vigorous effort has been made for the global elimination of *P. falciparum* malaria, but asymptomatic individuals are regarded as reservoirs for parasite transmission due to the lack of effective diagnostic methods for detecting the parasite in endemic areas.

For the treatment of malaria, the spread of anti-malarial drug resistance has become a major problem. If it is not handled appropriately, it could reverse the malaria control program and worldwide containment achieved so far. Therefore, more research is necessary to find new anti-malarial drugs for combating multidrug-resistant *P. falciparum* [253].

The factors like complex life cycle, genetic diversity, and the various immune escape mechanisms developed by these parasites, i.e., antigenic variations in *P. falciparum* [97], are the major obstacles responsible for the delay in the development of a suitable vaccine. Because of the high number of polymorphism or allele-specific variations in the proteins, single protein-based vaccines have limited success [254]. Vaccines composed of a chimeric molecule of the pre-erythrocytic target have shown protective responses in mice models [255]. The current efforts towards malaria parasite and vector control should be complemented with an effective *P. falciparum* vaccine for the successful fight against the malaria burden. Presently, vaccines from different stages of *P. falciparum* are in development and in the coming days, they might be combined (multivalent vaccines) to overcome the parasitic immune control mechanism attributed to antigenic polymorphism. A better understanding of the *P. falciparum* life cycle and the parasites’ interaction with the host should provide guidelines for the future vaccine development programs. The ideal malaria vaccine should be safe, highly effective, stable, easy to administer, and must provide long-term immunity. Such vaccines should also be cost-effective and affordable in poor malaria endemic areas.

Nowadays, researchers are fully aware of using suitable adjuvants for the development of novel vaccines. With regard to malaria, as mentioned above, an effective vaccine requires long-lasting primary and memory antigen-specific CD4^+^ T cell, B cell, and CTL responses. However, age-old adjuvants like alum, which skews the immune response to Th2, are not suitable for malaria vaccine development. Importantly, a combination of Th1 and Th2 adjuvants with a proper delivery system is needed for malaria vaccine development. Many prediction models have confirmed that activation of the IFN-γ signaling pathway is the best target to confer vaccine-induced protection against *P. falciparum* [256]. Next-generation vaccines should have wide spectrum activity against parasites, as some of the endemic residents encounter infection with more than one parasite simultaneously [257]. On top of that, vaccines should not induce suppressive Tregs, which are shown to aid in the parasite immune evasion mechanisms [110]. Further, an ideal adjuvant should be able to produce a protective immune response with a less frequent number of administrations. Nevertheless, we noticed that many adjuvants enhance effective immune responses with experimental antigens or model antigens, whereas a similar trend might not happen with the real antigens and unpaired antigens (Bonam, SR and Kumar HMS, personal observation; unpublished data). All the above details should help in the selection of specific adjuvant systems for the development of novel vaccines that elicit the highest humoral and cellular responses. Nowadays, analytical characterization of formulations to obtain vital information (i.e., particle size, morphology, and concentration of components, physicochemical interactions, stability, and others) are routinely performed. On the other hand, detailed studies are required to establish the biological interactions and determine the exact mechanism of action [201]. Systems vaccinology has helped considerably to uncover the unidentified vaccine adjuvant mechanism(s). Finally, the repository of novel Th1/Th2 adjuvants should be available to vaccine researchers to test their antigens for developing novel, promising malaria vaccines.

In summary, there is a need to coordinate research in multiple directions to develop an efficient malaria vaccine, such as (1) clear understanding of the life cycle of the malaria parasite, (2) identification and characterization of targets of interest with a broad range of strains covered, (3) identification of suitable adjuvant, which could enhance the magnitude and quality of the immunogenicity of antigen(s) without age factor [172], and (4) characterization and selection of a suitable ‘perfect mix’ (e.g., emulsion, liposomes, nanoparticles, etc.) for the delivery of the final vaccine to the host [173]. In essence, a detailed spatiotemporal analysis of parasite invasion biology will provide more information for effective targets. By tackling all these parameters, we could also support the poor responders in the vaccination.

## Figures and Tables

**Figure 1 vaccines-09-01072-f001:**
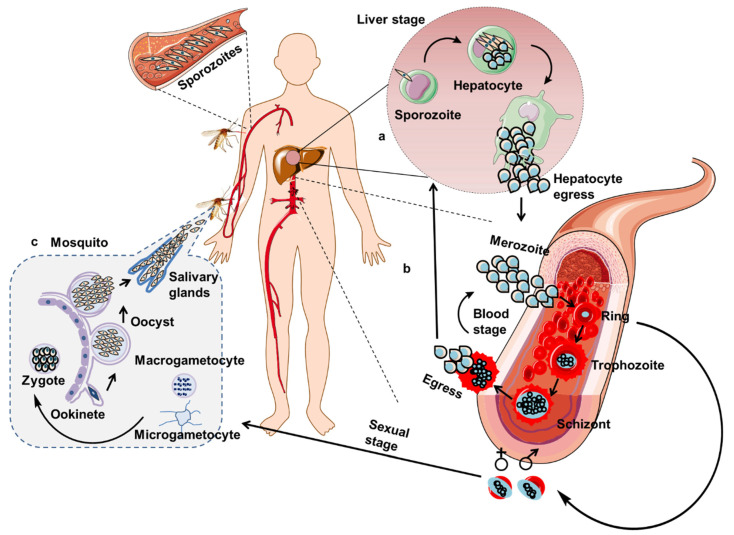
Complex life cycle of *P. falciparum*. The life cycle has 3 stages: the pre-erythrocytic and erythrocytic stages in humans (host) and sexual process in the mosquito vector. **a**|Pre-erythrocytic stage. **b**|Erythrocytic stage. **c**|Mosquito/sexual stage.

**Figure 2 vaccines-09-01072-f002:**
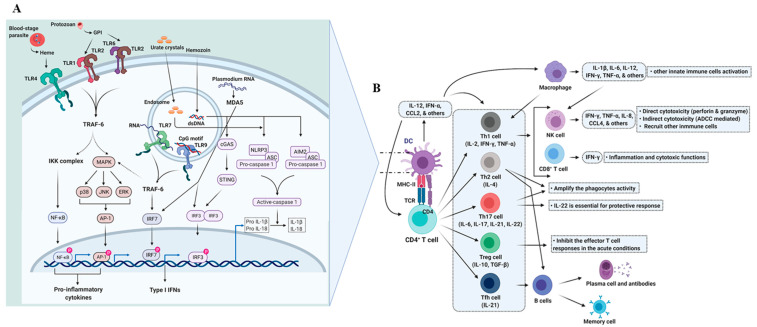
Immunity in malaria infection. **A|Activation of dendritic cells**. During the infection, both liver- and blood-stage parasites are recognized by the several PRRs present in the cells of the innate immune system. Majorly, monocytes, macrophages, and DCs have a significant role in the recognition. Vast literature has documented the TLR-mediated signals by parasite components, such as glycosylphosphatidylinositol (GPI), hemozoin, and DNA, upon infection. In addition, hemozoin and uric acid (released from the dying cells) activate the inflammasome. Furthermore, many other malaria parasite PAMPs are recognized by the different other unknown receptors and sensors. Once DCs/macrophages recognize the PAMPs and DAMPs, they phagocyte and process the pathogen, followed by the antigen presentation, to the T cells (reviewed in [25]). In addition, *Plasmodium* RNA (particularly in the hepatocytes) is sensed by the MDA5 ((RIG-I)-like receptor) and activates a type 1 IFN response via IRF3 and IRF7 signaling [26]. The above-mentioned innate immune responses help in the development of antigen-specific adaptive immunity. **B|Activation of adaptive immunity**. CD4^+^ T cells are activated by the DCs, and under the influence of cytokine milieu, CD4^+^ T cells are polarized into Th1, Th2, Th17, Treg, and Tfh cells. Each CD4^+^ T cell subset has a significant role in shaping the immune response (reviewed in [27]). NK cells perceive the signals by cytokines, which are produced from DCs, monocytes, and/or macrophages. Once activated, NK cells secrete inflammatory cytokines (IL-8, IFN-γ, TNF-α, CCL4, and others), which act as danger signals to gain other immune cells’ attention. NK cells also perform cytotoxicity as one of their main functions. NK cells either directly or indirectly kill the infected cells using perforin and granzyme and antibody-dependent cell cytotoxicity (ADCC), respectively. Antigen cross-presentation by DCs activates the CD8^+^ T cells, which produce IFN-γ upon activation. IFN-γ-producing CD8^+^ T cells perform inflammatory as well as cytotoxic (perforin and granzyme B mediated) functions. Though DCs induce a balanced immune response, the parasite could promote a severe pathology of cerebral malaria as a part of the immune evasion strategy [28].

**Figure 3 vaccines-09-01072-f003:**
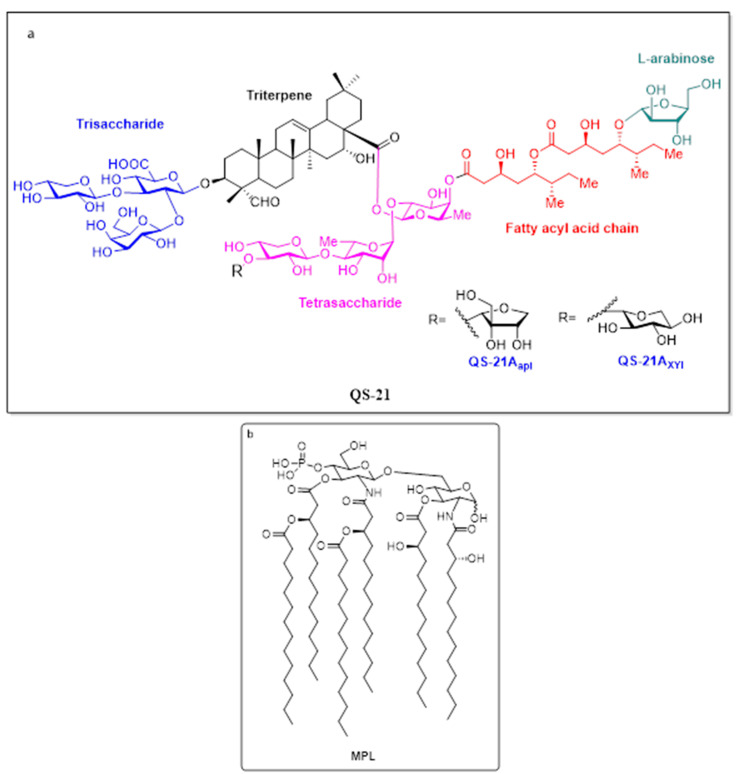
Structures of QS-21 (**a**) and MPL (**b**).

**Figure 4 vaccines-09-01072-f004:**
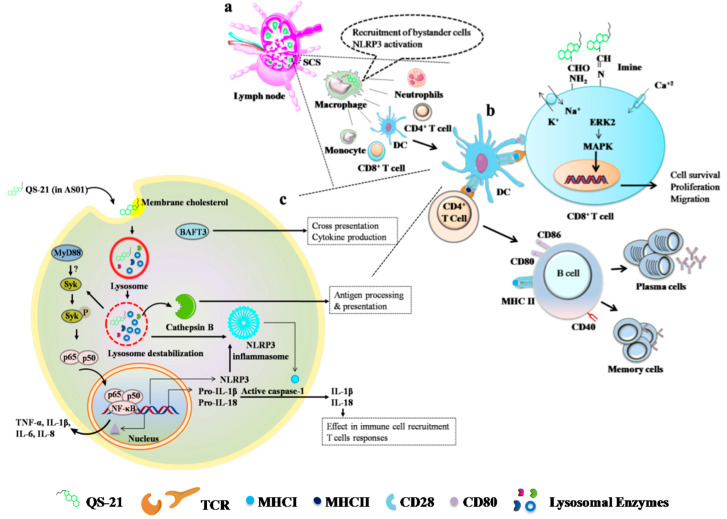
**Proposed mechanisms of action of QS-21 in AS01 and QS-21 alone**. **a**|Upon intramuscular injection, QS-21 (within the liposomes) drains to the lymph nodes and localizes at the subcapsular sinus (SCS) macrophages [240]. Macrophages activated via NLRP3 inflammasome pathways recruit other bystander cells including neutrophils, monocytes, and DCs [241]. **b**|QS-21 (within the liposomes) helps in the MHC-II and MHC-I antigen cross-presentation by forming a lipid body in the DCs and activating T cells [240,242]. Occasionally, in another way, QS-21 (alone) containing aldehyde moiety interacts with the T cell surface receptor (ε-amino acid) (e.g., CD2) and induces costimulatory signals [211]. These interactions, followed by ion exchanges (K^+^, Na^+^, Ca^2+^) and the activation of mitogen-activated protein (MAP) kinase, ERK2, and transcription factors, lead to cytokine release and cell proliferation, differentiation, and migration. **c**|QS-21 (within the liposomes) enters DCs via cholesterol-mediated endocytosis and situates itself in the lysosomes, where it destabilizes the lysosomes by forming size-specific pores. Cathepsin B (cysteine proteases) released from the lysosomes has an effective role in the immune functions mediated by QS-21. QS-21 (alone or in liposomes) activates the inflammasome (NLRP3), like other adjuvants, which induces active caspase 1 production. Activated caspase 1 coverts the pro-inflammatory cytokines (IL-1β and IL-18) into active forms [240].

**Table 1 vaccines-09-01072-t001:** List of vaccines under clinical testing without specific vaccine adjuvants.

Malaria Vaccine	Clinical Trial Identifier	Current Stage
ChAd63 RH5 (chimpanzee adenovirus serotype 63 reticulocyte-binding protein homolog 5)	NCT02181088	Phase 1
MVA (modified vaccinia virus Ankara) RH5	NCT02181088	Phase 1
PEBS-POC1 (synthetic protein containing 131 amino acids)	NCT01605786	Phase 1
ChAd63-METRAP (multiple epitope string and thrombospondin-related adhesion protein)	NCT03084289	Phase 1
MVA METRAP	NCT03084289	Phase 1
DNA-Ad (contains a combination of circumsporozoite (CS) protein and AMA1)	NCT00870987	Phase 2
PfSPZ (*P. falciparum* (Pf) sporozoite (SPZ))	NCT02601716	Phase 2
Ad35.CS.01 (*P. falciparum* CS surface antigen is inserted in a replication deficient Adenovirus 35 backbone)	NCT01018459	Phase 1
CS protein expressed either in MVA, or an attenuated Fowl pox virus strain (FP9).	NCT00121771	Phase 1
AdCh63-MSP1 (merozoite surface protein-1) and MVA-MSP1	NCT01003314	Phase 2
GMZ2 (recombinant hybrid of the glutamate rich protein (GLURP) and the merozoite surface protein 3 (MSP 3))	NCT00424944	Phase 1
FP9-PP and MVA-PP (FP9 polyprotein, modified virus Ankara polyprotein)	NCT00374998	Phase 1
p52-p36-GAP (genetically attenuated parasite malaria vaccine)	NCT01024686	Phase 2
PfSPZ-GA1 (genetically attenuated PfSPZ)	NCT03163121	Phase 1
ChAdOx1 LS2 (malaria liver-stage dual antigen LS2 (LSA1 and LSAP2) fused with the transmembrane domain from shark invariant chain) and MVA LS2	NCT03203421	Phase 2/Phase 1
DNA-Ad (it contains a liver-stage antigen (circumsporozoite protein) and an antigen (apical membrane antigen 1))	NCT00870987	Phase 1
NMRC-M3V-Ad-PfCA (NMRC + multi-antigen multi-stage, malaria vaccine + adenovectored + *P. falciparum* CSP and AMA1 antigens), is a combination of two recombinant adenovirus-derived constructs (adenovectors)	NCT00392015	Phase 1

**Table 2 vaccines-09-01072-t002:** Status of malarial vaccines with vaccine adjuvants ^#^.

Adjuvant	Vaccine	Life-Cycle Stage	Clinical Trial Identifier	Current Stage
Aluminium hydroxide/Alhydrogel^®^	Lyophilized PEBS synthetic protein (PfPEBS) (synthetic protein containing 131 amino acids)	Pre-erythrocytic and blood stage	NCT01605786	Phase 2
	AMA1-C1 (combination of the 3D7 and FVO alleles of *P. falciparum* apical membrane antigen-1 (AMA1))	Blood stage	NCT00984763 NCT00114010	Phase 2 Phase 1
	PRIMVAC (VAR2CSA protein)	Blood stage	NCT02658253	Phase 1
	P27A protein	Blood stage	NCT01949909	Phase 1
	Pfs25-EPA (Pfs25 has been conjugated to *Pseudomonas aeruginosa* ExoProtein A) (EPA)	Sexual stage	NCT01434381	Phase 1
	Pfs25 VLP	Sexual stage	NCT02013687	Phase 1
	AMA1-DiCo	Blood stage	NCT02014727	Phase 1
	Pfs25M-EPA, Pfs230D1M-EPA	Sexual stage	NCT02334462	Phase 1
	AdCh63 AMA1 + MVA AMA1 + AMA1-C1	Blood stage	NCT01351948	Phase 1
	BSAM-2 (mixture of two proteins found on the surface of merozoites, AMA1 and MSP1 (42))	Blood stage	NCT00889616	Phase 1
	ICC-1132	Pre-erythrocytic stage	NCT00587249	Phase 1
	PAMVAC	Blood stage	NCT02647489	Phase 1
	MSP3-LSP	Blood stage	NCT01341704	Phase 2
	MSP1 42-C1	Blood stage	NCT00320658	Phase 1
Aluminum phosphate	Erythrocyte-binding antigen 175 kDA region II-non-glycosylated (EBA-175 RII-NG)	Blood stage	NCT01026246	Phase 1
AS01	RH5.1 (protein ectodomain of the PfRH5 (amino acids E26—Q526) antigen)	Blood stage	NCT02927145	Phase 2
	Pfs25M-EPA, Pfs230D1M-EPA (PfS230D1M conjugated to *Pseudomonas aeruginosa* ExoProtein A (EPA))	Sexual stage	NCT02942277	Phase 1
AS01B	R21 (RTS, S-like vaccine)	Pre-erythrocytic stage	NCT02600975	Phase 1
	FMP012 (Escherichia coli-expressed *P. falciparum* cell-traversal protein for ookinetes and sporozoites (PfCelTOS))	Sexual stage	NCT02174978	Phase 1
	*P. falciparum* malaria protein (FMP)010	Blood stage	NCT00666380	Phase 1
	FMP2.1 (AMA1 malaria antigen)	Blood stage	NCT00385047	Phase 1
AS01E	RTS, S	Pre-erythrocytic stage	NCT00380393	Phase 2
AS02A	Falciparum Merozoite Protein-1 (FMP1)	Blood stage	NCT00308061	Phase 1
	FMP2.1 (AMA1 malaria antigen)	Blood stage	NCT00385047	Phase 2
	Falciparum Malaria Protein 11	Pre-erythrocytic stage	NCT00312702	Phase 2
AS02D	RTS, S	Pre-erythrocytic stage	NCT00289185	Phase 2
ODN 2006 (7909)	MSP1 42-C1	Blood stage	NCT00320658	Phase 1
	AMA1-C1 (combination of the 3D7 and FVO alleles of *P. falciparum* apical membrane antigen-1 (AMA1))	Blood stage	NCT00984763	Phase 2
	AdCh63 AMA1 + MVA AMA1 + AMA1-C1	Blood stage	NCT01351948	Phase 1
	BSAM-2	Blood stage	NCT00889616	Phase 1
Glucopyranosyl Lipid Adjuvant-Liposome-QS-21 Formulation (GLA-LSQ)	PAMVAC	Blood stage	NCT02647489	Phase 1
	Recombinant circumsporozoite protein (rCSP) malaria vaccine administered with and without AP 10-602 (GLA-LSQ)	Pre-erythrocytic stage	NCT03589794	Phase 1
Glucopyranosyl Lipid Adjuvant-Stable Emulsion (GLA-SE)	PAMVAC	Blood stage	NCT02647489	Phase 1
	PRIMVAC (VAR2CSA protein)	Blood stage	NCT02658253	Phase 1
	P27A protein	Blood stage	NCT01949909	Phase 1
	AMA1-DiCo	Blood stage	NCT02014727	Phase 1
	FMP012	Sexual stage	NCT01540474	Phase 1
Matrix-M1 (a bifunctional matrix protein of influenza virus)	R21	Pre-erythrocytic stage	NCT02925403	Phase 1
Montanide ISA51	PpPfs25	Sexual stage	NCT00295581	Phase 1
Montanide ISA 720	PfCS102 (antigen of the sporozoite protein)	Pre-erythrocytic stage	NCT01031524	Phase 1
Virosomes	PEV301 & 302 (it includes two antigens (CSP and AMA1-derived))	Pre-erythrocytic stage and blood stage	NCT00513669	Phase 1

Abbreviations: AMA1-DiCo, apical membrane antigen 1 diversity covering; ICC-1132, a vaccine candidate composed of *P. falciparum* hepatitis B virus core particle, which contains T- and B-cell epitopes from the repeat region of the C terminus circumsporozoite protein; METRAP, multiple-epitope thrombospondin-related adhesion protein; MSP3-LSP, merozoite surface protein-3 long synthetic peptide; PfCelTOS, *P. falciparum* cell traversal protein of ookinetes and sporozoites; PfAMA1, *P. falciparum* apical membrane antigen 1; PfPEBS, *P. falciparum* pre-erythrocytic and blood stage; PfMSP1, *P. falciparum* merozoite surface protein 1; PfRH5, *P. falciparum* reticulocyte-binding protein homolog 5; PfSERA5, *P. falciparum* Serine Repeat Antigen-5; P27A protein, unstructured 104 mer synthetic peptide from *P. falciparum* trophozoite exported protein 1; VAR2CSA, variant surface antigen 2CSA; VLP, virus-like particles. ^#^ We searched for malarial vaccines in clinical trials website (https://clinicaltrials.gov/ accessed on 1 June 2020) and selected the studies pertinent to *P. falciparum*. Many vaccines use alhydrogel as adjuvant, however, at different doses and different conditions. To our knowledge, many vaccines with a similar type of antigens using a diversified combination of adjuvants have been under clinical trials, which are not listed here.

## Data Availability

Not applicable.

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
