# Peer review of "Plasmodium falciparum Malaria Vaccines and Vaccine Adjuvants"

_vaccines, 2021, doi:10.3390/vaccines9101072_

Round 1
Reviewer 1 Report
I do not have any main criticism. There are some minor, mostly formal shortcomings what are easy to correct. They are listed in below.
line 207, 210, 214, 286 -IRBC. The abbreviation „iRBC” is used generally in the text (defined in line 339). Please, define the abbreviation in line 207.
line 336 – “can i) prevent or merozoites”, it seems that the word “or” is unnecessary
line 339 – “and iv) promote”, change for “and v) promote”
line 340 – “antibody-cell-dependent inhibition (ADCI) [106]. In ADCI”, It would be more logic to use ACDI as abbreviation.
lines 348-349 – “the binding to iRBC to iRBC”, delete one of the expressions of “to iRBC”
line 387 - “have h helped”, eliminate the “h”
line 1458 - „Cerevisiae” change for „cerevisiae”
Table 2 – “pseudomonas aeruginosa”, replace with “Pseudomonas aeruginosa”
References
2 – Add url: https://www.who.int/news-room/feature-stories/detail/world-malaria-report-2019
4 - Add url: https://www.who.int/news/item/19-11-2018-this-year-s-world-malaria-report-at-a-glance
15 - Add url: https://apps.who.int/iris/bitstream/handle/10665/162441/9789241549127_eng.pdf
10, 63, 64 – Use italic letters for Plasmodium yoelii
12, 13, 16, 40, 41, 44, 47, 74, 84, 89, 105, 107, 108, 111, 112, 120, 121, 128, 139, 142-144, 153, 182, 184, 188, 208-210, 215, 220, 222-224, 244 –Use italic letters for Plasmodium falciparum
32, 33, 36, 37, 59, 67, 77, 116 –Use italic letters for Plasmodium
35, 58, 97, 122 –Use italic letters for Plasmodium berghei
102 –Use italic letters for Plasmodium vivax and Plasmodium cynomolgi
104, 140, 229 - Use italic letters for Plasmodium vivax
108 -Delete from doi „%J Journal of Experimental Medicine”
110 - Delete from doi „Doi”; Use italic letters for stevor, rif and Plasmodium falciparum
119 - Use italic letters for Plasmodium falciparum and Plasmodium vivax
Author Response
To
Prof. Dr. Ralph A. Tripp,
Editor-in-Chief,
Vaccines
Dear Prof. Ralph,
We would like to thank for considering our work for the reviewing process. We have considered the worthy reviewers’ suggestions and improved the manuscript. The revisions in the revised manuscript are differentiated with track changes. Brief answerers are provided in the present rebuttal letter. When we say line numbers, they represent the revised manuscript. We hope that the current manuscript is acceptable for publication in your esteemed journal.
Thank you very much for your consideration.
Sincerely yours,
Reviewer 1:
I do not have any main criticism. There are some minor, mostly formal shortcomings what are easy to correct. They are listed in below.
Thank you very much for consideration.
line 207, 210, 214, 286 -IRBC. The abbreviation „iRBC” is used generally in the text (defined in line 339). Please, define the abbreviation in line 207.
Fixed
line 336 – “can i) prevent or merozoites”, it seems that the word “or” is unnecessary
Fixed
line 339 – “and iv) promote”, change for “and v) promote”
Fixed
line 340 – “antibody-cell-dependent inhibition (ADCI) [106]. In ADCI”, It would be more logic to use ACDI as abbreviation.
Fixed
lines 348-349 – “the binding to iRBC to iRBC”, delete one of the expressions of “to iRBC”
Fixed
line 387 - “have h helped”, eliminate the “h”
Fixed
line 1458 - „Cerevisiae” change for „cerevisiae”
Fixed
Table 2 – “pseudomonas aeruginosa”, replace with “Pseudomonas aeruginosa”
Fixed
References
2 – Add url: https://www.who.int/news-room/feature-stories/detail/world-malaria-report-2019
4 - Add url: https://www.who.int/news/item/19-11-2018-this-year-s-world-malaria-report-at-a-glance
15 - Add url: https://apps.who.int/iris/bitstream/handle/10665/162441/9789241549127_eng.pdf
10, 63, 64 – Use italic letters for Plasmodium yoelii
12, 13, 16, 40, 41, 44, 47, 74, 84, 89, 105, 107, 108, 111, 112, 120, 121, 128, 139, 142-144, 153, 182, 184, 188, 208-210, 215, 220, 222-224, 244 –Use italic letters for Plasmodium falciparum
32, 33, 36, 37, 59, 67, 77, 116 –Use italic letters for Plasmodium
35, 58, 97, 122 –Use italic letters for Plasmodium berghei
102 –Use italic letters for Plasmodium vivax and Plasmodium cynomolgi
104, 140, 229 - Use italic letters for Plasmodium vivax
108 -Delete from doi „%J Journal of Experimental Medicine”
110 - Delete from doi „Doi”; Use italic letters for stevor, rif and Plasmodium falciparum
119 - Use italic letters for Plasmodium falciparum and Plasmodium vivax
As suggested, all the references are formatted in the revised manuscript.

Reviewer 2 Report
The clear strength of this review is its discussion of cellular immune responses to malaria; other sections were less authoritative and less focussed.
A further strength is the series of point-form sections (particularly Box 1 and Outstanding Questions) but there is an almost complete disconnect between the main text and these sections. This would be a much stronger review if it focussed on elaborating the points raised in these sections.
Section 2 is titled "Immunity against P. falciparum", but it relies heavily on data derived from other plasmodium sp. This is, of course, to some extent unavoidable, but it is important that it is clear to the reader which findings have experimental support in the human system, and which are known only for rodent or other models. Although some parts of the text manage this well, other sections (eg 2.1) could be improved.
In contrast to the detailed discussion of innate and T-cell immunity, the treatment of antibody-mediated immunity (2.2.4) is brief, relatively superficial, and fails to adequately describe the importance of antibodies for both naturally- and vaccine-derived immunity to P. falciparum.
Figure legends throughout are overly long, and present material that does not help the reader to understand the figure, but rather that either repeats what is already in the main text, or which should be moved to the main text.
lines 424-425: This is an oft-stated but superficial claim that is very difficult to relate to malaria, where immunity to both intracellular and extracellular forms are important
There are numerous examples of grammatically incorrect or imprecise and sometimes misleading language. A limited set of examples are listed below, but the entire ms needs careful revision.
line 27: "clinically approved vaccine adjuvants". Vaccines are assessed for approval as defined formulations, and approval is granted as such. It is therefore incorrect to suggest that an adjuvant, per se, is approved.
line 82: "infections can be managed through vector control approaches". Transmission can be controlled through vector-targeting approaches, but clearly management of infection demands parasite- or patient-directed interventions.
line 101: "clinical immunity and sterile immunity are acquired". There is no evidence that sterile immunity to P. falciparum can be acquired.
line 104: 'i.e.' should be 'such as'
Fig 2 lacks panel labels (A, B) and text is too small to be read at normal magnification
line 138: "protecting malaria induced pathology" should be "protecting from ..."
line 148: "which majorly caused by the parasite immune evasion strategies". Grammar.
line 298: "induction of a strong immune response may not be desirable". While there may be a need for a well-calibrated T-cell response, the need for a very strong (though well-targeted) B-cell response is clear.
Author Response
To
Prof. Dr. Ralph A. Tripp,
Editor-in-Chief,
Vaccines
Dear Prof. Ralph,
We would like to thank for considering our work for the reviewing process. We have considered the worthy reviewers’ suggestions and improved the manuscript. The revisions in the revised manuscript are differentiated with track changes. Brief answerers are provided in the present rebuttal letter. When we say line numbers, they represent the revised manuscript. We hope that the current manuscript is acceptable for publication in your esteemed journal.
Thank you very much for your consideration.
Sincerely yours,
Reviewer 2:
The clear strength of this review is its discussion of cellular immune responses to malaria; other sections were less authoritative and less focussed. A further strength is the series of point-form sections (particularly Box 1 and Outstanding Questions) but there is an almost complete disconnect between the main text and these sections. This would be a much stronger review if it focussed on elaborating the points raised in these sections.
Thank you very much for the suggestion. The possible improvement are made in the revised manuscript.
Section 2 is titled "Immunity against P. falciparum", but it relies heavily on data derived from other plasmodium sp. This is, of course, to some extent unavoidable, but it is important that it is clear to the reader which findings have experimental support in the human system, and which are known only for rodent or other models. Although some parts of the text manage this well, other sections (eg 2.1) could be improved.
The section 2.1 was improved by taking the reviewers suggestion.
In contrast to the detailed discussion of innate and T-cell immunity, the treatment of antibody-mediated immunity (2.2.4) is brief, relatively superficial, and fails to adequately describe the importance of antibodies for both naturally- and vaccine-derived immunity to P. falciparum.
The revised manuscript contains additional paragraphs about “antibody-mediated immunity”.
Figure legends throughout are overly long, and present material that does not help the reader to understand the figure, but rather that either repeats what is already in the main text, or which should be moved to the main text.
As per the suggestion, the important text was clubbed with the main text and repeated text was deleted.
lines 424-425: This is an oft-stated but superficial claim that is very difficult to relate to malaria, where immunity to both intracellular and extracellular forms are important
Modified
There are numerous examples of grammatically incorrect or imprecise and sometimes misleading language. A limited set of examples are listed below, but the entire ms needs careful revision.
The manuscript was grammatically corrected.
line 27: "clinically approved vaccine adjuvants". Vaccines are assessed for approval as defined formulations, and approval is granted as such. It is therefore incorrect to suggest that an adjuvant, per se, is approved.
The word “approved” is deleted.
line 82: "infections can be managed through vector control approaches". Transmission can be controlled through vector-targeting approaches, but clearly management of infection demands parasite- or patient-directed interventions.
Fixed
line 101: "clinical immunity and sterile immunity are acquired". There is no evidence that sterile immunity to P. falciparum can be acquired.
The word “sterile immunity” is deleted.
line 104: 'i.e.' should be 'such as'
Fixed
Fig 2 lacks panel labels (A, B) and text is too small to be read at normal magnification
The present figure has labels and the text size in the figure was increased.
line 138: "protecting malaria induced pathology" should be "protecting from ..."
The sentence has modified and moved to the main text.
line 148: "which majorly caused by the parasite immune evasion strategies". Grammar.
Fixed
line 298: "induction of a strong immune response may not be desirable". While there may be a need for a well-calibrated T-cell response, the need for a very strong (though well-targeted) B-cell response is clear.
A new sentence was added.